

# Cytogenetic screening of chromosomal abnormalities and genetic analysis of FSH receptor Ala307Thr and Ser680Asn genes in amenorrheic patients

Bushra A. Kanaan[1], Mushtak T.S. Al-Ouqaili[1] and Rafal M. Murshid[2]

[1] Department of Microbiology, University of Anbar, College of Medicine, Ramadi, Al-Anbar Province, Iraq
[2] Department of Gynecology and Obstetrics, University of Anbar, College of Medicine, Ramadi, Al-Anbar Province, Iraq

Corresponding author
Mushtak T.S. Al-Ouqaili,
ph.dr.mushtak_72@uoanbar.edu.iq

## ABSTRACT

**Objective**. Amenorrhea is a rare reproductive medical condition defined by the absence of menstruation during puberty or later life. This study aims to establish the frequency and pattern of chromosomal abnormalities (CA) in both primary amenorrhea (PA) and secondary amenorrhea (SA), and further to detect the genetic changes in exon 10 at nucleotide positions 919 and 2039 of the genotypes Thr307Ala, and Asn680Ser, respectively.

**Design, settings and patients**. This cross-sectional study was conducted on a sample of seventy amenorrhoeic women according to the Helsinki declaration rules of medical ethics, as divided into 40 (57.14%) with PA and 30 (42.86%) with SA, and 30 healthy women with normal menstruation as the control. The chromosomal karyotyping was performed according to the ISCN, 2020. PCR products were submitted to RFLP and Sanger sequencing for women with normal karyotype and high FSH serum levels.

**Results**. The classical Turner Syndrome was the most common CA in PA, followed by isochromosome X [46, Xi(X)(q10)], mosaicism of Turner and isochromosome X [45, X /46, Xi(X)(q10)], sex reversal (46, XY) and (46, XX,-3,+der3,-19,del 19 p). Abnormal SA cases were characterized by mosaicism Turner syndrome (45,X/46,XX) and (46,XX,-3,+der3,X,+derX). The homozygous genotypes AA and GG of Ala307Thr (rs6165) in the FSHR gene are most common in PA, while the homozygous genotype AA is more common in SA. GG and AG genotypes of Ser680Asn (rs6166) are more frequent in Iraqi patients with PA and SA compared to the healthy control women. Both PCR-RFLP and Sanger sequencing indicated a marked matching between genotypes.

**Conclusions**. The study emphasizes the need for cytogenetic analysis to determine the genetic basis of PA and SA. Further, genotyping for women with normal karyotype and high FSH serum concentrations via PCR-RFLP should be considered for the precise diagnosis and development of appropriate management of and counselling for these patients.

## INTRODUCTION

The absence of a menstrual period for one month or longer is known as amenorrhea, and is one of the most typical gynecological issues amongst adolescent girls and older women Amenorrhea is the sixth-most common cause of female infertility, according to the World Health Organization, which estimates that 15% of the world's population is infertile (*Susheela, 2019*). About 2–5% of women of childbearing age experience primary or secondary amenorrhea (*Al-Jaroudi et al., 2019*).

Primary amenorrhea (PA) and secondary amenorrhea (SA) are the two distinct forms of amenorrhea (SA). PA is diagnosed when a woman has not reached menarche by age 13 in those who have not yet developed secondary sexual characteristics, or by age 15 in those who have (*Ghosh, Roy & Halder, 2020*). According to statistics from the Ministry of Health for 2017 and 2018, 9.68% and 17.78% of women in Iraq experience problems with menarche and PA, respectively (*Nijeeb, Alkazaz & Yaseen, 2020*).

Most cases of PA are caused by anatomical abnormalities, high levels of follicle-stimulating hormone (FSH), hyperprolactinemia, hypothalamic amenorrhea, polycystic ovary syndrome (PCOS), or gonadal dysgenesis, which most frequently result in Turner syndrome (TS). Chromosome abnormalities are present in 16% to 64% of cases of primary amenorrhea (*Korgaonkar et al., 2019*; *Klein, Paradise & Reeder, 2019*).

SA is the three- or six-month absence of regular or irregular menstruation (*Ansari, Chatur & Walode, 2021*). SA may have a variety of causes such as pregnancy, lactation, thyroid issues, hyperprolactinemia, hyperandrogenism (including polycystic ovarian syndrome), hypogonadotropic hypogonadism (hypothalamic-pituitary dysfunction), and endometrial suppression by hormonal birth control are examples of conditions that can be attributed to hormones; structural causes include damage to the endometrium (Asherman syndrome) and obstruction of the outflow tract (cervical stenosis) (*Martini et al., 2016*). However, chromosomal abnormalities like monosomy X, trisomy of chromosome XXX, or Mosaic Turner syndrome 45, XO/46, XX cause the majority of cases of SA (*Hughes, 2008*).

Granulosa cells in females express the FSHR gene, which controls estrogen production, Graafian follicle maturation, and granulosa cell proliferation (*Chrusciel et al., 2019*). FSHR spans more than 190K bases and is found at chromosome 2.p21. A G protein-coupled receptor called the human FSHR has a long extracellular domain (ECD), a 7 transmembrane domain (7TMD) with three short intracellular loops and three additional loops, and an intracellular tail (*Banerjee, Joseph & Mahale, 2021*).

Various single nucleotide polymorphisms of the FSHR gene have been identified in various populations; however, the two polymorphisms located in exon ten at nucleotide positions 919 (Thr307Ala) and 2039 (Asn680Ser) are the focus of extensive research to determine the FSHR response to FSH stimulation. A mutant allele of these polymorphisms associated with the lower response of receptor and is significantly connected with the high level of basal FSH (*Achrekar et al., 2010*; *Panghiyangani et al., 2019*). This occurs because the cells in the ovaries are less sensitive to follicle-stimulating hormone, which are responsible for stimulating the ovaries to generate hormones of negative sex-steroid feedback. Hypergonadotropic hypogonadism can disrupt your menstrual cycle by interfering with

egg maturation and ovulation, resulting in missed periods (a condition called amenorrhea) (*Hoffman, 2016*).

The chromosomal abnormalities place a heavy financial, social, and health burden on patients and their families. To assess risk and provide genetic counselling, cytogenetic and molecular diagnoses are crucial. Thus, this study has been undertaken to determine the frequency and pattern of chromosomal abnormalities in PA and SA in Iraq. Further, to detect the course of treatment or warning such patients about early menopause in cases of Turner syndrome, the likelihood of infertility in cases of mosaic Turner. Furthermore, to help the clinician in announce the recommendation to undertake hormone therapy, and the risk of gonadal malignancy in cases of XY dysgenesis in case of sex reversal.

## PATIENTS AND METHODS

### Selection of study patients

This cross-sectional study was conducted from February to August 2022 on the one hundred women included in this study who were separated into three groups as follows: (I) is composed of 40 (40.0%) women who suffer from PA; (II) consisted of 30 (30.0%) women with SA; while group (III) represented 30 (30.0%) healthy control women with normal menstruation cycles (Fig. 1). All women with amenorrhea, either PA or SA, who were referred to the Department of Obstetrics and Gynecology in Ramadi Teaching Hospital for Child and Maternity and other private clinics in Baghdad city, were included in the study. The expert clinician made the initial diagnosis of amenorrhea based on the available medical history, physical examination, hormonal profile, and ultrasonography.

Inclusion criteria for PA were females characterized by the absence of secondary sexual characteristics or females with an age of more than 16 years with normal growth, development, and the appearance of secondary sexual characteristics (pubic and axillary hair and breast development). Regarding SA, the assigned criteria include those with no menses cycle during three periods or over six months. The exclusion criteria included women showing abnormal hormonal statuses, such as autoimmune thyroid disorders or prolactin imbalance, in addition to infections, and exposure to radiation.

The normal karyotype females suffering from PA ($n = 18$) and SA ($n = 15$) with a high level of FSH hormone >20 mIU/ml were submitted to molecular study of the FSH receptor gene mutation and polymorphism.

### Ethics statement

According to the Helsinki Declaration, this study received approval from the University of Anbar's Medical Ethics Committee on February 13, 2022, in Ramadi, Iraq (approval number 8). All study participants provided their written consent.

### Collection of samples

#### Blood sample for serological study

Two milliliters of peripheral blood samples were collected using a sterile disposable syringe, drawn in a 6 ml gel and clot activator tube, Spain. After allowing it to clot in the water bath (37 °C). The sera were separated *via* centrifugation for 5 min at 1500 rpm to study FSH

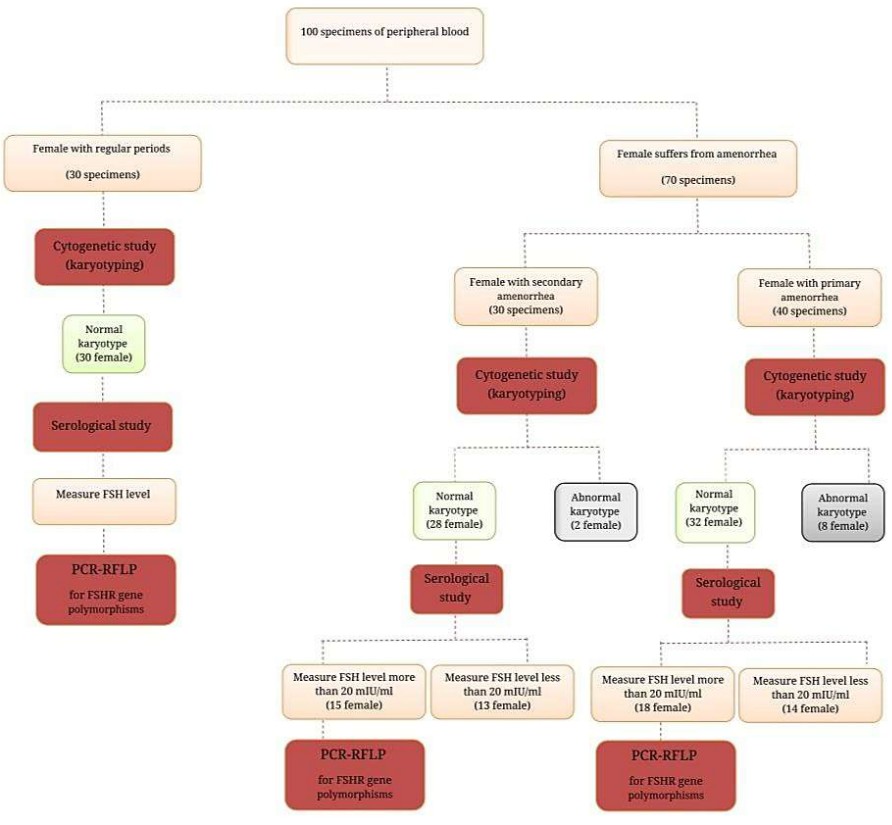

**Figure 1  Ideogram of study design.**

and LH, as performed *via* a TOSOH Bioscience (AIA-1800) system automated by taking 100 μl of serum.

### Blood sample for cytogenetic study

Two milliliters of peripheral blood samples were taken from women with PA or SA using a sterile disposable syringe and injected into a lithium heparin tube. The blood was promptly cultured and analyzed for chromosomal abnormalities.

### Blood sample for molecular study

For patients with normal karyotyping, another blood aspirate for two milliliters of blood into EDTA tubes was taken to determine FSH receptor gene mutation and polymorphism.

## Serology part of the study

All participants submitted to evaluation for hormonal studies, including FSH and LH hormone, in human serum on a TOSOH Automated Immunoassay System (TOSOH Bioscience, Tokyo, Japan).

## Cytogenetic study
### Karyotyping

Lymphocyte cultures were generated according to the standard procedure described by Chiecchio, Ganguly, Khamees and Al-Ouqaili (*Chiecchio et al., 2006*; *Kanaan, Al-Ouqaili & Murshed, 2022*; *Khamees & Al-Ouqaili, 2022*). The amount of 1,500 µl of peripheral blood samples was aseptically transferred into sterile culture tubes with four milliliters of RPMI-1640 medium solution (HiMedia, Mumbai, India). Into each centrifuged conical tube, one milliliter of fetal bovine serum (Biowest, Nuaillé, France), 200 µl Phytohemagglutinin (Biowest, European), and antibiotics (60 µl ampicillin and 60 µl streptomycin) were added.

The culture medium was kept at −20 °C until needed. The culture tubes were appropriately marked before being incubated in an incubator for 72 h. After 71 h, each culture tube received 100 µl of the anti-metaphase drug colchicine (Biowest, European) to arrest the cells. The cell suspensions underwent 15 min, 1500 rpm centrifugation after 30 min of incubation. After the supernatant was discarded, the pellet was treated with a gentle drop-by-drop application of hypotonic solution (0.075 M KCl). The centrifuged tubes were then incubated for 20 min at 37 °C. Once more, the tubes were carefully centrifuged at 1500 rpm for 15 min. Then, the supernatant was removed, and seven milliliters of freshly prepared fixation solution (3:1 methanol to glacial acetic acid) was added to the pellet and mixed thoroughly. The tubes were washed with fresh fixative solution 3–4 times. The supernatant was then removed following the completion of centrifugation, and this step repeated until a clear pellet was obtained and stored the pellet with two milliliters of fixation solution at −20 °C.

### Slide preparation

Pre-cleaned slides were refrigerated (4 °C) in triple-distilled water for 2–3 h (*Lekha et al., 2020*). The cell culture mixture was well mixed by vortex mixer, and 3–4 drops of the cell suspension were dropped uniformly from an acceptable distance (30–50 cm) at a 45-degree angle onto oil-free slides, which were then allowed to dry at room temperature before being labeled. The slides were stained with Giemsa stain. Giemsa stain powder weighing two grams and 100 ml of absolute methanol were combined to make the appropriate solution. The solution was stirred for three hours using a Hotplate Magnetic stirrer model L-81 with a magnetic bar at room temperature, and then filtered using a Millipore sterile syringe filter membrane of 0.45 µm and stored in a tightly closed dark bottle. One milliliter of the stock solution is mixed with four milliliters of Sorenson's buffer for immediate staining (*Ali & Salih, 2021*).

### Karyotype analysis

The MetaClass Karyotyping apparatus (Microptics S.L., Barcelona, Spain) and a fluorescence microscope were used to determine the chromosomal status (Euromex, Arnhem, Netherland). Standard cytogenetic protocols examined metaphases and chromosome preparations from peripheral blood cultures. G-banding was used for cytogenetic analysis. All participants had twenty metaphases examined. In some cases, the analysis was expanded to 50 metaphases, particularly when abnormalities and mosaicism

were suspected. The International System of Human Cytogenetic Nomenclature was used to report chromosomal abnormalities (*McGowan-Jordan, Hastings & Moore, 2020*)

## Molecular part of the study
### DNA extraction and quantification
A SaMag blood DNA extraction kit was employed with the SaMag-12 automatic nucleic acid extraction system (Sacace biotechnologies, Italy) for extraction of genomic DNA from 400 µl human whole blood, according to the Al-Qaysi protocols (*Al-Qaysi, Al-Ouqaili & Al-Meani, 2020*).  The blood DNA extraction includes the following steps: lysis, binding, washing, and elution. Quantus™ Fluorometer with QuantiFluor® dsDNA System (Promega, Madison, WI, USA) was used to detect the concentration of extracted DNA. DNA concentrations were measured (*Al-Ouqaili, Al-Taei & Al-Najjar, 2018*). DNA purity was also calculated using an OD260/OD280 UV spectrophotometer (*Al-Qaysi, Al-Ouqaili & Al-Meani, 2020*).

### PCR amplification
The requested primers were purchased from Alpha DNA Company (Canada). The primers were dissolved and lyophilized in nuclease-free water to make a stock solution to reach a final concentration of 1,000 pmol/l. Polymerase Chain Reaction (PCR), ESCO, Riverside, CA, USA, was used to amplify DNA fragments. The template DNA and primers were added to an AccuPower® PCR PreMix tube requested from Bioneer, Daejeon, Republic of Korea. The PCR reaction was carried out in a 20 µL mixture containing 1 U Top DNA polymerase, 250 µM each: dNTP (dATP, dCTP, dGTP, dTTP), 1X reaction buffer with 1.5 mM MgCl2, 0 Stabilizer and tracking dye, amplification primers (2 µL, each), target DNA (5 µL), and nuclease-free water (11 µL). To amplify the FSHR gene using the following PCR condition, each gene was programmed to perform 30 cycles of initial denaturation (5 min at 94 °C), denaturation (30 s. at 94 °C), annealing (30 s. at 55 °C) for Ala307Thr or (30 s. at 60 °C) for Ser680Asn, extension (1 min. at 72 °C) and a final extension step (5 min. at 72 °C), in addition to holding the temperature at 4 °C. Thereafter, five µl of PCR product were added to each 1.5% agarose gel well. The electrodes were attached to the power supply at 50 V for 5 min, then 100 V for 60 min. DNA bands were visualized using an ultraviolet transilluminator (Vilber Lourmat, Lemont, IL, USA). The PCR products with sizes of 577 bp (Ala307Thr) and 520 bp (Ser680Asn) were investigated (Table 1).

### Restriction Fragment Length Polymorphism (RFLP)
RFLP analysis of the Ala307Thr (https://www.ncbi.nlm.nih.gov/snp/rs6165) variant: The PCR amplification product of this gene of FSHR (577 bp) was digested with AhdI restriction enzyme (BioLabs-USA). AhdI digestion was performed at 37 °C for 3 h, during inactivation at 65 °C for 20 min., and visualized on 1.5% agarose gel stained with ethidium bromide in 1X TBE buffer solution at a voltage of 100 V for 1 h.

### RFLP analysis of the Ser680Asn (rs6166) variant
The PCR product of this gene of FSHR (520 bp) was digested with the BsrI restriction enzyme (BioLabs-USA). BsrI digestion was performed at 65 °C for 3 h, then inactivation

**Table 1  Primers and restriction enzymes used in this study.**

| Polymorphisms | Primer's sequence | Product size | Restriction enzymes | Allele size | Reference |
|---|---|---|---|---|---|
| Ala307Thr | F: 5′CCTGCACAAAGACAGTGATG-3′; | 577 bp | AhdI | Ala:403+174 | *Sundblad et al. (2004)* |
|  | R: 5′TGGCAAAGACAGTGAAAAG-3′ |  |  | Thr:403+143+31 |  |
| Ser680Asn | F: 5′TTTGTGGTCATCTGTGGCTGC-3′; | 520 bp | BsrI | Asn:520 | *Sudo et al. (2002)* |
|  | R:5′CAAAGGCAAGGACTGAATTATCATT-3′ |  |  | Ser:413+107 |  |

at 80 °C for 20 min. Enzyme-digested PCR products were evaluated *via* electrophoresis in 1.5% agarose gels with ethidium bromide in 1X TBE buffer solution at 100 V for 1 h.

## Sequencing

The Sanger dideoxynucleoside sequencing technique using ABI3730XL by Macrogen Corporation (Seoul, South Korea) was performed to detect any polymorphisms in the samples of this study. The sequences were then run through the standard gene BLAST (Basic Local Alignment Search Tool) program, which is available at the National Center Biotechnology Information (NCBI) online at (http://www.ncbi.nlm.nih.gov), and sequences were aligned using the genius software.

## Statistical analysis

Statistical analysis was performed using SPSS software version 22(Armonk, NY, USA). An Independent $t$-test was used to compare the studied groups. $P$-value of 0.05 was considered the significance level. The WINPEPI computer program (version 11.63) was used to estimate the statistical significance of the $P$-values calculated using Fisher's exact test, and the odds ratio was assessed by a special $\chi^2$ formula.

## RESULTS

The age category of patients with PA was ranged from 14 to 25 years with a mean of 17.82 ± 2.934. In SA, the age range was from 18 to 36 years with a mean of 26.266 ± 5.037. In healthy subjects (control), it was from 14–33 years, with a mean of 22.5 ± 5.04. In patients with PA, the FSH values were 47.689 ± 37.25 and LH 14.07 ± 13.67. Also, body mass index (BMI) was a 24.17 ± 4.35. The values in SA patients were 43.86 ± 40.44, 12.85 ± 11.752, and 6.253 ± 4.228 for FSH, LH, and BMI, respectively The statistical analysis of FSH and LH levels revealed that there were significant statistical differences between female study patients and healthy female subjects ($P$-value less than 0.05). All related details are reported in Table 2.

### The cytogenetic part in patients with primary and early secondary amenorrhea

The chromosomal analysis of women with PA and SA was performed on 70 cases and 30 cases with women with a normal menstruation cycle as the control The chromosomal analysis revealing that the karyotype was typical (46,XX) in healthy women and in 60 cases (85.7% of all cases with amenorrhea). However, chromosomal aberration was present in 8 (20.0%) cases with PA and in 2 cases (6.6%) with SA.

**Table 2** Comparison of follicle-stimulating and Luteinizing hormones in females with primary and secondary amenorrhea, with healthy females in the control group.

| Parameters | Study Cases/ Control | No. | Mean | Std. Deviation | P- Value Sig. (two-tailed) |
|---|---|---|---|---|---|
| FSH | Primary amenorrhea cases | 40 | 47.689 | 37.255 | 0.000 |
| | Control women | 30 | 5.879 | 0.957 | |
| | Secondary amenorrhea cases | 30 | 43.865 | 40.441 | 0.000 |
| | Control women | 30 | 5.879 | 0.957 | |
| LH | Primary amenorrhea cases | 40 | 14.078 | 13.67 | 0.01 |
| | Control women | 30 | 5.224 | 1.855 | |
| | Secondary amenorrhea cases | 30 | 12.853 | 11.752 | 0.01 |
| | Control women | 30 | 5.224 | 1.855 | |

### Primary amenorrhea

Classical Turner Syndrome (X chromosome monosomy 45, X) was the most common CA (Fig. 2), where 4 (50.0%) cases followed by one case (12.5%) of Isochromosome X [46, Xi(X)(q10)] (Fig. 3), one case (12.5%) of mosaicism of Turner and isochromosome X [45, X /46, Xi(X)(q10)], one case (12.5%) of sex reversal [46, XY] (Fig. 4), and the final case (12.5%) of the deletion of chromosome19p and derivative in chromosomes 3 [46, XX,-3,+der3,-19,del 19 p], as revealed in Table 3.

In comparison results of this study with other studies conducted in Iran, India, and Turkey, there are converging and diverging ratios according to the type of study, epidemiological region, and study population size, as shown in Table 4

### Secondary amenorrhea

In a study of women with SA of 30 females, 28 (93.4%) had a normal karyotype (46, XX), whilst 2 females (6.6%) had abnormal karyotypes (Table 5). These women had Mosaicism Turner syndrome (45, X/46, XX) (Fig. 5) and an abnormal structural karyotype (46,XX,-3,+der3, X,+derX).

In compares the percentage of different karyotypes in patients with SA with the findings of other studies were reflected in Table 6.

## The molecular part of the study
### Association between Ala307Thr (rs6165) gene polymorphism and amenorrhea

The PCR product of the Ala307Thr (rs6165) gene of FSHR on Exon 10 was 577 bp. Results for the amplified DNA fragment of this gene are shown in Fig. 6.

Then the amplified DNA was analyzed *via* PCR–RFLP, the PCR product of which was incubated with the restriction enzymes (AhdI), with digestion performed at 37 °C for 3 h, inactivation at 65 °C for 20 min, and which was then visualized on 1.5% agarose gel stained with ethidium bromide. The results revealed three genotypes (AA, AG, and GG) (Fig. 7).

The distribution of genotypes and alleles in PA cases is shown in Table 7. The study control group populations revealed that AG is the common genotype, while in patients' groups it is AA. However, the frequency distribution of homozygous genotype AA and

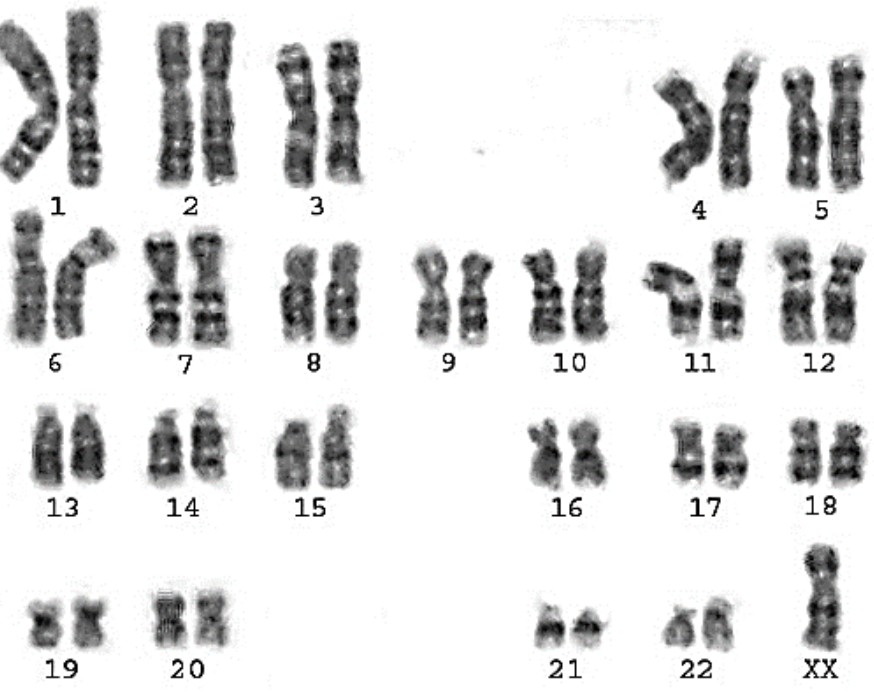

**Figure 2** Karyotype of pure turner monosomy X: 45,X.

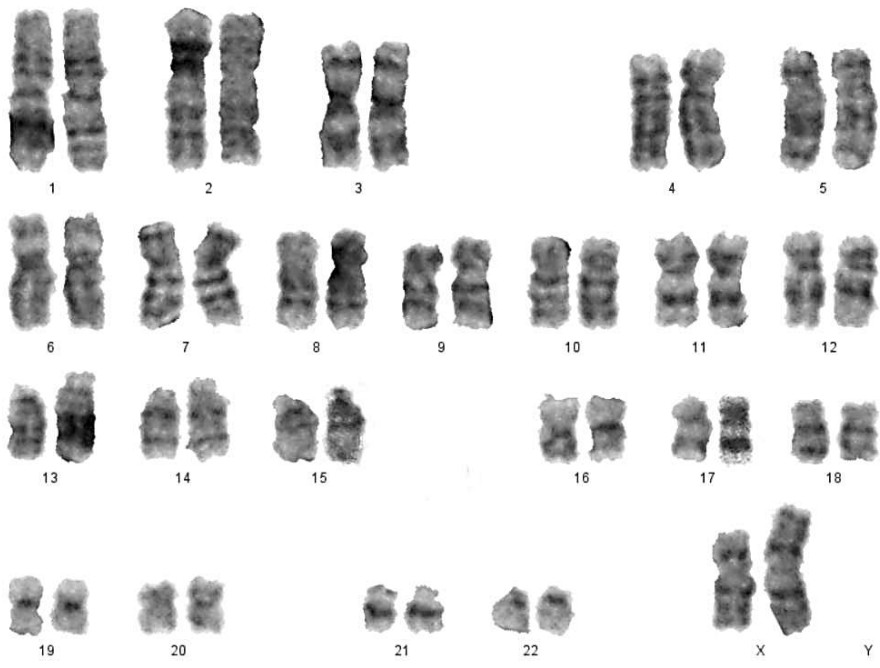

**Figure 3** Karyotype of isochromosome of long arm of X: 46,Xi (Xq).

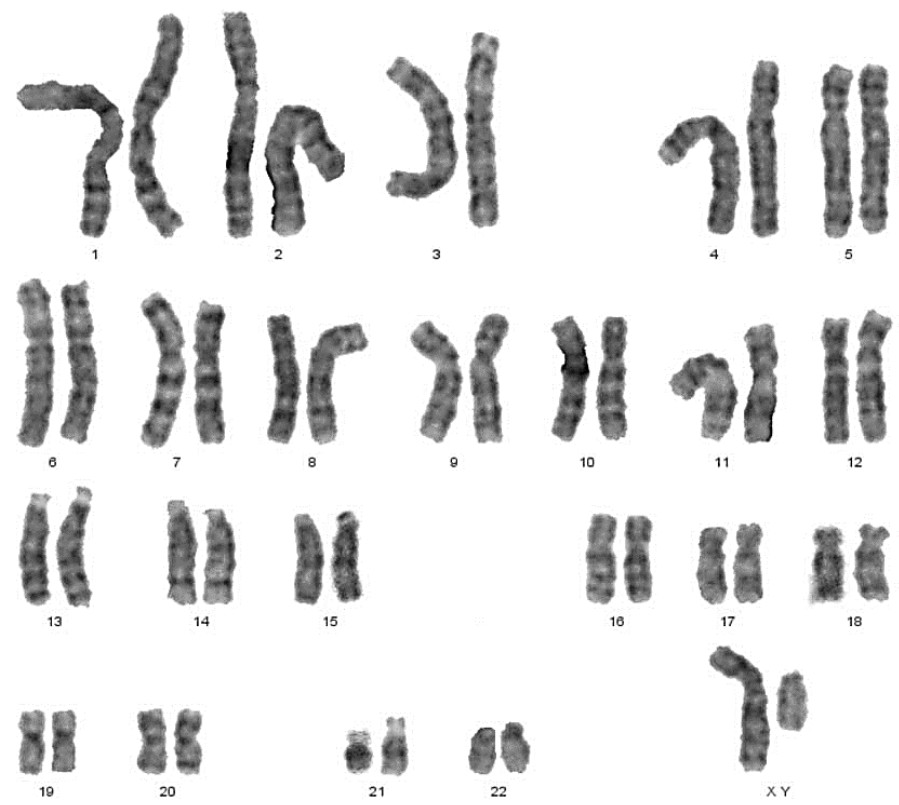

**Figure 4** Sex reversal karyotype: 46,XY.

GG differed significantly between the control group and PA subjects. The homozygous genotype AA in PA cases was 50.0%, while in healthy women was 16.67%, showing a positive association with PA (odds ratio 5.00; CI 95% [1.27–19.93]); the significant genotype homozygous GG was present in 11.11% in PA cases and 40.0% in the control group, showing a negative association with PA (odds ratio 0.19; CI 95% [0.03–0.92]). Additionally, the frequency distributions of the heterozygous genotype AG in PA cases and the control group were 38.89% and 43.33%, respectively.

In contrast, the odds ratio for allele A was 3.66, indicating a positive association with the disease in terms of the odds ratio, which could be considered a risk allele (odds ratio >1.0), which could be considered an etiological fraction that may result in Iraqi women being more susceptible to amenorrhea, while the odds ratio for allele G was 0.27, indicating a negative association with the disease. These results were in agreement with other studies on Iraqi women with PA, in which AA genotypes were higher in patients than in the controls, and allele A was a risk factor associated with amenorrhea.

For SA, the distribution of genotypes and alleles can be seen in Table 8. The frequency distribution of homozygous genotype AA in SA was higher than in the control group (53.3% *vs.* 16.7%) in SA and healthy women with normal cycles, respectively. The significant homozygous genotype AA positively affects SA (odds ratio 5.71; CI 95% [1.35–24.16]). No significant heterozygous genotype AG or homozygous genotype GG were revealed in

**Table 3  Cytogenetic variation, karyotype, and frequency of patients with primary amenorrhea.**

| Cytogenetic variation | Karyotype | No. of cases | Frequency (%) | Hormonal status | Ultrasonography findings |
|---|---|---|---|---|---|
| **Normal Karyotype** | 46, XX | 32 | 80.0% | Increased FSH (18), Decreased FSH (14) | Normal (9), Small size of ovaries and uterus (12), small ovaries (7), pre-puberty (4) |
| **Abnormal Karyotype:** | | 8 | 20.0% | | |
| **A-Monosomy X** | 45, X | 4 | 50.0% | Increased FSH | Streak ovaries and a normal uterus (2), Streak ovaries and a small uterus (1), small, atrophied ovaries with infantile uterus (1) (fig 2). |
| **B-Isochromosome X** | 46,Xi(X)(q10) | 1 | 12.5% | Increased FSH/LH | Absent ovaries with infantile uterus (fig 3). |
| **C-Mosaicism of Turner and isochromosome X** | 45,X /46,Xi(X)(q10) | 1 | 12.5% | Increased FSH/LH | Streak ovaries and an infantile uterus (fig 3). |
| **D-Sex reversal** | 46,XY | 1 | 12.5% | Increased FSH/LH | Streak gonads with small size of uterus (fig 3). |
| **E-Other** | 46,XX,-3,+der3,-19,del 19 p | 1 | 12.5% | Decreased FSH/LH | Small size of the uterus, and the uterus. |

**Table 4  Chromosomal aberration frequency of primary amenorrhea as reported in various studies.**

| Results | Present study | *Soltani, Mirzaei & Ayatollahi (2021)* | *Lekha et al. (2020)* | *Korgaonkar et al. (2019)* | *Ghosh et al. (2018)* | *Geckinli et al. (2014)* |
|---|---|---|---|---|---|---|
| **Study population** | Iraq | Iran | India | India | India | Turkey |
| **Total number of cases** | 40 | 200 | 53 | 490 | 150 | 175 |
| **Frequency of abnormal karyotype (%)** | 20% ($n = 8$) | 29% ($n = 58$) | 32.08% (n ($n = 17$) | 24.7% ($n = 121$) | 24% ($n = 36$) | 25% ($n = 44$) |
| **Numerical abnormality (%)** | 50% ($n = 4$) | 50% ($n = 29$) | 64.7% ($n = 11$) | 43.8% ($n = 53$) | 50% ($n = 18$) | 34% ($n = 15$) |
| **Structural abnormality (%)** | 37.5% ($n = 3$) | 32.8% ($n = 19$) | 23.5% ($n = 4$) | 26.4% ($n = 32$) | 41.67% ($n = 15$) | 31.8% ($n = 14$) |
| **Male karyotype** | 12.5% ($n = 1$) | 17.2% ($n = 10$) | 11.8% ($n = 2$) | 29.8% ($n = 36$) | 8.33% ($n = 3$) | 29.5% |

the comparison between SA and healthy women($P$-value greater than 0.05) was observed, indicating a negative association with SA (odds ratio 0.48; CI 95% [0.11–1.85] and 0.38; CI 95% [0.07–1.60], respectively).

The interpretation of the PCR-RFLP results showed that the odds ratio for allele A was 3.22, so the association with the SA according to the odds ratio could be considered

**Table 5  Cytogenetic variation, karyotype, and frequency of patients with secondary amenorrhea.**

| Cytogenetic variation | Karyotype | No. of cases | Frequency (%) | Hormonal status | Ultrasonography findings |
|---|---|---|---|---|---|
| **Normal Karyotype** | 46,XX | 28 | 93.4% | Increased FSH (15), Decreased FSH (13) | Normal in size and shape for both ovaries and uterus (15), small size of ovaries and normal uterus (9), normal ovaries with anteverted uterus (3), normal ovaries with bicornuate uterus (1). |
| **Abnormal karyotype** | | 2 | 6.6% | | |
| **A-Mosaicism Turner syndrome** | 45,X/46,XX | 1 | 3.3% | Increased FSH/LH | Small size of ovaries with uterus size and shape were normal (Fig. 5). |
| **B-Sex chromosome with a structural abnormality** | 46,XX,-3,+der3,X,+derX. | 1 | 3.3% | Decreased FSH/LH | Streak ovaries with small size of uterus. |

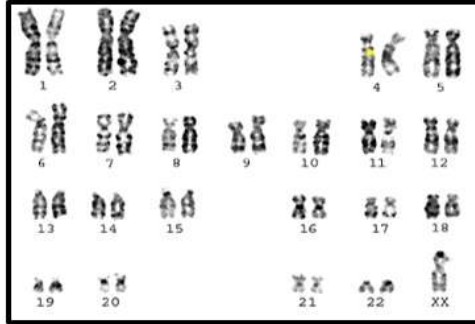 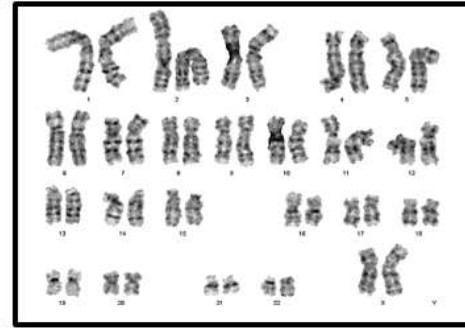

**Figure 5  Karyotype of monosomy X (45,X) and (46,XX).**

a risk allele. In contrast, the odds ratio for the G allele was 0.31, which is interpreted as a negative association with amenorrhea, which could be considered an etiological agent in Iraqi patients and may need a larger sample size to gain a clear understanding of the relationship.

This is a technique that reveals only specific single nucleotide polymorphism (SNP). After that, DNA sequencing was performed on 10 specimens that underwent RFLP. The sequencing technique was observed to give a wide view of the study region, at which the sequences yielded were analyzed by NCBI blast online. The sequencing results confirmed the results obtained by the PCR-RFLP for the same gene, Ala307Thr(rs6165), in nucleotide position 919. The three genotypes in the chromatogram sequencing are shown in Fig. 8.

**Table 6  Frequency of chromosomal aberration in secondary amenorrhea as reported in various studies.**

| Results | Present study | Bai et al. (2022) | Afshar et al. (2016) | Leelavathy, Lakshmikantha & Sayee (2015) | Safai et al. (2012) | Butnariu et al. (2011) |
|---|---|---|---|---|---|---|
| Study population | Iraq | Indian | Iran | Indian | Iran | Romania |
| Total number of cases | 30 | 49 | 24 | 231 | 94 | 38 |
| Frequency of abnormal karyotype (%) | 6.6%(n = 2) | 18%(n = 9) | 8.3%(n = 2) | 16%(n = 37) | 5.3%(n = 5) | 18.4%(n = 7) |
| Numerical abnormality (%) | 50%(n = 1) | 22.2%(n = 2) | 100%(n = 2) | 43%(n = 16) | 40%(n = 2) | 100%(n = 7) |
| Structural abnormality (%) | 50%(n = 1) | 77.8%(n = 7) | – | 35%(n = 13) | 60%(n = 3) | – |
| XY female and variants (%) | – | – | – | 22%(n = 8) | – | – |

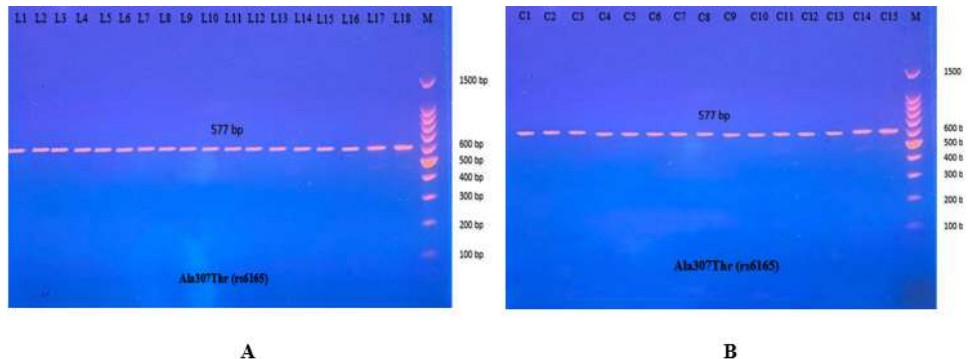

**Figure 6  PCR product of FSHR gene (Ala307Thr, 577 bp).** The product was subject to electrophoresis on 1.5% agarose. (A) PCR product of specimen with patients. (B) PCR product of control specimen.

**Table 7  Distribution of Ala307Thr (rs6165) polymorphism genotypes in primary amenorrhea cases and healthy females as control.**

| Genotypes | Primary amenorrhea (n = 18) | Controls (n = 30) | P-Value | Odds ratio | CI95% |
|---|---|---|---|---|---|
| | | Ala307Thr (rs6165) | | | |
| Thr/Thr (AA) | 9 (50.0%) | 5 (16.67%) | 0.015 | 5.00 | 1.27 to19.93 |
| Ala/Thr (AG) | 7 (38.89%) | 13 (43.33%) | 0.886 | 0.83 | 0.24 to 2.80 |
| Ala/Ala (GG) | 2 (11.11%) | 12 (40.0%) | 0.036 | 0.19 | 0.03 to 0.92 |
| | | Alleles distribution | | | |
| A | 25(69.44%) | 23(38.33%) | 0.004 | 3.66 | 1.51 to 8.98 |
| G | 11(30.56%) | 37(61.67%) | 0.004 | 0.27 | 0.11 to 0.66 |
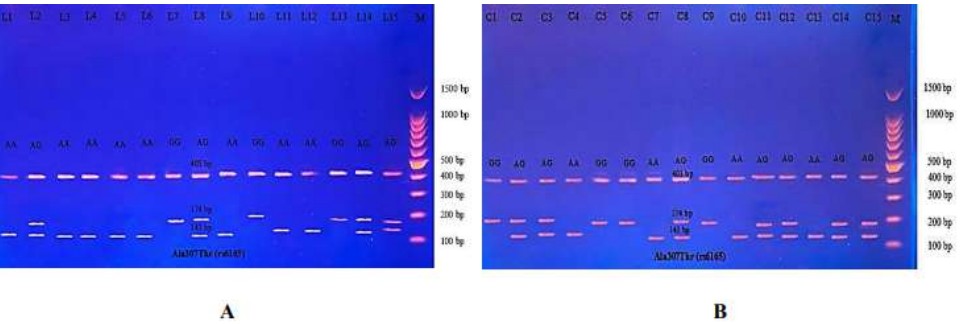

**Figure 7  Electrophoretogram of DNA fragments for Ala307Thr polymorphisms after digestion with AhdI restriction enzymes.** (A) PCR-RFLP results of patients; (B) PCR-RFLP results with normal menstruation cycle. Homozygote A/A was shown by the bands of 403 bp, 143 bp and 31 bp. Homozygote G/G was shown by the 403 bp and 174 bp bands. Heterozygote A/G was shown by the bands of 403 bp, 174 bp, 143 bp and 31 bp. Note that the 31 bp bands were run off the gel.

**Table 8  Frequency distribution of Ala307Thr (rs6165) polymorphism genotypes in secondary amenorrhea cases and healthy females as control.**

| Genotypes | Secondary amenorrhea ($n = 15$) | Controls ($n = 30$) | P-Value | Odds ratio | CI95% |
|---|---|---|---|---|---|
| | | **Ala307Thr (rs6165)** | | | |
| Thr/Thr (AA) | 8 (53.3%) | 5 (16.7%) | 0.010 | 5.71 | 1.35 to 24.16 |
| Ala/Thr (AG) | 4 (26.7%) | 13 (43.3%) | 0.267 | 0.48 | 0.11 to 1.85 |
| Ala/Ala (GG) | 3 (20.0%) | 12 (40.0%) | 0.257 | 0.38 | 0.07 to 1.60 |
| | | **Alleles distribution** | | | |
| A | 20 (66.67%) | 23 (38.33%) | 0.011 | 3.22 | 1.27 to 8.26 |
| G | 10 (33.33%) | 37 (61.67%) | 0.011 | 0.31 | 0.12 to 0.79 |

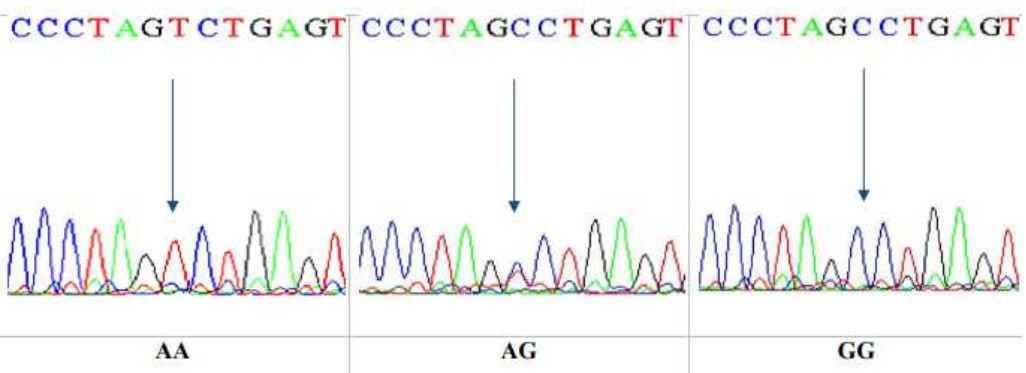

**Figure 8  DNA sequencing of the forward strand of Ala307Thr polymorphisms (AA, AG and GG).**

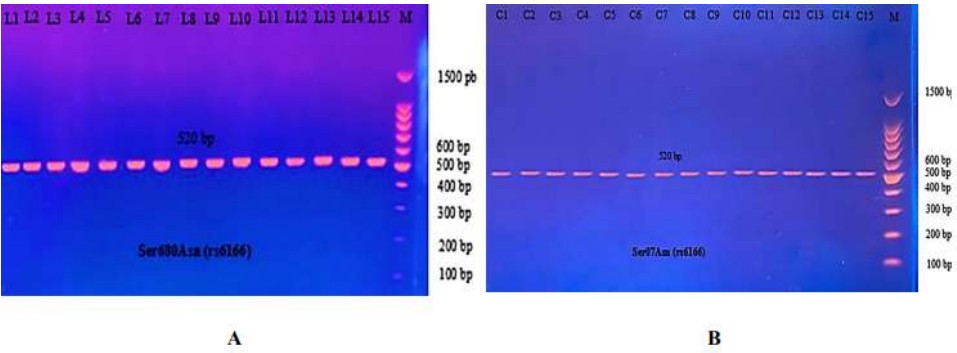

**Figure 9** **The PCR product of Ser680Asn polymorphism gene on exon 10 of FSHR was 520 bp.** (A) PCR product of patients. (B) PCR product of control specimen.

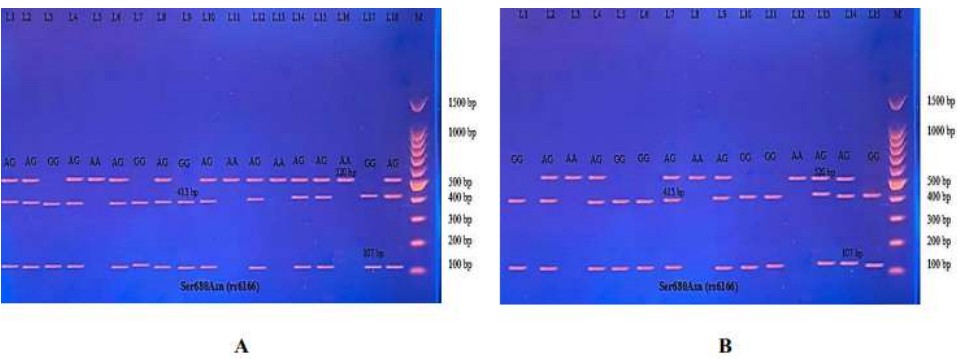

**Figure 10** **(A–B) Electrophoretogram of DNA fragments for Ser680Asn polymorphisms after digestion with BsrI restriction enzymes.** Homozygote A/A was indicated by the bands at 520 bp. Homozygote G/G was indicated by the bands at 413 bp and 107 bp. Heterozygote A/G was indicated by the bands at 520 bp, 413 bp, and 107 bp.

### Association between Ser680Asn (rs6166) gene polymorphism and amenorrhea

The most common studies regarding FSHR revealed location 680, which switches serine to asparagine (Ser680Asn). The PCR product of this gene of FSHR on Exon 10 was 520 bp, as seen in Fig. 9.

The PCR product samples were incubated with the digestive BsrI Restriction Enzyme for 3 h and run on 1.5% agarose gel electrophoresis for 60 min, 100 V. The results revealed three genotypes: AA, AG, and GG. The band at 520 bp showed Homozygote A/A. Homozygote G/G was inferred from the 413 bp and 107 bp bands. Heterozygote A/G was inferred by the bands at 520 bp, 413 bp, and 107 bp (Fig. 10).

The results reported in Table 9 show the distribution of the three genotypes of Ser680Asn polymorphism in PA cases and in the control women. Significant differences were found in the genotype's distribution and allele frequency between the PA patients and the control groups. Significant homozygous genotype AA shows a negative association with the

**Table 9 Frequency distribution of Ser680Asn (rs6166) polymorphism genotypes in primary amenorrhea cases and control group.**

| Genotypes | Primary amenorrhea (n = 18) | Controls (n = 30) | P-Value | Odds ratio | CI95% |
|---|---|---|---|---|---|
| | | Ser680Asn (rs6166) | | | |
| Asn/Asn (AA) | 4 (22.22%) | 16 (53.33%) | 0.028 | 0.25 | 0.06 to 0.94 |
| Ser/Asn (AG) | 10 (55.56%) | 11 (36.67%) | 0.187 | 2.16 | 0.64 to 7.30 |
| Ser/Ser (GG) | 4 (22.22%) | 3 (10.0%) | 0.316 | 2.57 | 0.46 to 15.20 |
| | | Alleles distribution | | | |
| A | 18(50.00%) | 43(71.67%) | 0.039 | 0.40 | 0.17 to 0.95 |
| G | 18(50.00%) | 17(28.33%) | 0.039 | 2.53 | 1.05 to 6.04 |

**Table 10 Frequency distribution of the Ser680Asn (rs6166) polymorphism in secondary amenorrhea and control group.**

| Genotypes | Secondary amenorrhea (n = 15) | Controls (n = 30) | P- Value | Odds ratio | CI95% |
|---|---|---|---|---|---|
| | | Ser680Asn (rs6166) | | | |
| Asn/Asn (AA) | 3 (20.0%) | 16 (53.33%) | 0.041 | 0.22 | 0.04 to 0.93 |
| Ser/Asn (AG) | 6 (40.0%) | 11 (36.67%) | 0.876 | 1.15 | 0.30 to 4.20 |
| Ser/Ser (GG) | 6 (40.0%) | 3 (10.0%) | 0.031 | 6.00 | 1.18 to 33.23 |
| | | Alleles distribution | | | |
| A | 12(40.00%) | 43(71.67%) | 0.004 | 0.26 | 0.10 to 0.67 |
| G | 18(60.00% | 17(28.33%) | 0.004 | 3.79 | 1.49 to 9.66 |

disease (odds ratio 0.25; CI95% between 0.06−0.94). No significance was revealed in the heterozygous AG and homozygous GG genotypes, but which showed a positive association with the disease (odds ratio 2.16; CI95% between 0.64−7.30 and 2.57; CI95% between 0.46–15.20, respectively).

On the other hand, the odds ratio for allele A was 0.4 with a CI95% between 0.17−0.95. Thus, a negative association with the disease, according to the odds ratio, could be considered a protective allele. Also, the odds ratio for allele G was 2.53 with a CI95% between 1.05−6.04, indicating a positive association with the disease, which could be considered an etiological agent that may make Iraqi women more susceptible to the disease.

In SA, the distribution of genotypes and alleles can be seen in Table 10. The frequency distribution of AA genotypes in the control group was higher than in SA cases (20.0% *vs.* 53.33% in SA and healthy women with normal cycles, respectively). The negative homozygous AA genotype association with the control (odds ratio 0.22 and CI95% between 0.04−0.93) with statistically significant (*P*-value = 0.041), while the GG genotypes revealed a positive association (odds ratio 6.00; CI95 from 1.18–33.23, *P*-value = 0.031). Further, a positive association between SA for the heterozygous AG genotype and the control was observed, for which the odds ratio was 1.15 and the CI95% was from 0.30 to 4.20, and no significant was noted (*P*-value = 0.876).

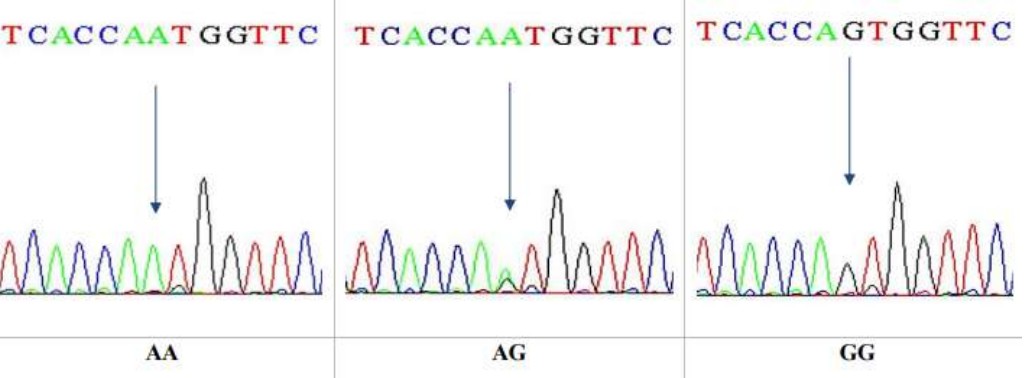

**Figure 11  DNA sequencing of the forward strand of Ser680Asn polymorphisms (AA, AG, and GG).**

On the other hand, a negative allele A association with SA (Odds ratio 0.26; CI95% from 0.10−0.67) is considered a protective allele. The positive allele G association (odds ratio 3.79; CI95% from 1.49−9.66) indicated a risk allele.

DNA sequencing results confirmed the data obtained from the PCR-RFLP as having the Ala307Thr rs6165. Sequencing chromatograms for the three genotypes AA, AG, and GG are shown in Fig. 11.

## DISCUSSION

Amenorrhea is a condition whose causes range from pregnancy, to the absence of the uterus and vagina, to hormonal imbalance, an excess of male testosterone, endometritis, unfavorable ovarian function, and Mullerian agenesis. Chromosomal abnormalities are an important factor in amenorrhea, according to cytogenetic studies. Chromosomal aberration of primary amenorrhea reporting rates ranges from 15.9% to 63.3%, and secondary amenorrhea reporting rates range from 3.8% to 44.4% (*Demirhan et al., 2014*).

In PA cases, 20% revealed a variety of CA. Of these, numerical chromosomal abnormalities were identified in 50%. Further, structural abnormalities were identified in 37.5%, and the sex reversal karyotype was detected in 12.5%.

A previous study by *Soltani, Mirzaei & Ayatollahi (2021)* in the northeast of Iran reported that 58 (29%) cases of study women with PA yielded different CAs. Out of 29 (50.0%) of patients with numerical aberrations, structural abnormalities were detected in 19 (32.8%) and pure sex reversal karyotypes in ten (17.2%). On the other hand, *Ghosh et al. (2018)* concluded that, for their study of Indian women, 36 (23.9%) patients suffered from PA that was found to be caused by CA. Of these, 18 cases (50%) had numerical abnormalities while 15 (41.67%) had structural chromosomal abnormalities; a further three cases (8.33%) showed sex reversal karyotype. Both of the above study results were in agreement with those observed with our study.

In an Indian study by *Lekha et al. (2020)*, 53 cases were distributed into 17 (32.08%) with CA, 11 (64.7%) with abnormal numerical karyotype, four (23.5%) were suffering from structural abnormalities, while two (11.8%) showed sex reversal karyotype. This

study indicated that numerical abnormalities had a higher frequency and sex reversal a lower frequency, in agreement with the current research.

In another Indian study by *Korgaonkar et al. (2019)*, 121 (24.7%) cases were identified with abnormal karyotypes. Numerical chromosomal abnormalities were detected in 53 (43.8%) cases, and structural abnormalities revealed 32 (26.4%) and 36 (29.8%) cases of sex reversal karyotype. Another previous study by *Geckinli et al. (2014)* in Turkey revealed 44 (25%) with CA, of which there were 15 (34%) numerical abnormalities, 14 (31.8%) structural abnormalities, and 13(29.5%) sex reversal karyotype. Our results do not entirely agree with these studies because the frequency of sex reversal karyotype in this study's region was greater than ours.

In this study, the majority of chromosomal abnormalities were observed in patients with PA due to Monosomy of chromosome X, constituting four cases (50%) of chromosomal abnormalities, where all of these patients had clinical symptoms of TS. Short stature and delayed puberty, and lack of incidence of secondary sexual characteristics were significant clinical symptoms amongst these patients. This result was in line with previous studies, which reported that TS is the most commonly observed CA in PA, and also strengthened the role of the sex chromosome in the reproduction of females (*Ali et al., 2018*; *Lekha et al., 2020*; *Soltani, Mirzaei & Ayatollahi, 2021*).

In this study, out of 30 cases of SA, two (6.6%) revealed a variety of CA. Of these, each numerical and structural abnormality was identified in 50% of the relevant sample. A previous study by *Afshar et al. (2016)* in Iran revealed that two cases (8.3%) of women with SA had chromosomal abnormalities, with both displaying clinical symptoms of TS. Additionally, a study by *Safai et al. (2012)* from Iran included 94 women with SA. This study determined five females (5.3%), two of whom(40%) had numerical abnormalities, and three (60%) structural abnormalities. These studies are in agreement with the frequency distribution of CA in SA with those observed in our study.

On the other hand, *Bai et al. (2022)* reported that nine cases (18%) of their study of women in India with SA yielded different CA. Two cases (22.2%) with SA patients with numerical aberration and structural abnormalities were detected (77.8%). Additionally, a study by *Leelavathy, Lakshmikantha & Sayee (2015)* in India revealed 37 women (16%) with abnormal karyotypes, out of whom 16 (43%) had numerical abnormalities, 13 (35%) structural abnormalities, and eight (22%) were XY females and variants. These studies indicate a higher frequency in comparison with the observations in this research.

In another study of SA completed in Romania by *Butnariu et al. (2011)* found seven females (18.4%) with abnormal karyotypes. All cases were with numerical abnormalities. In contrast with our study, half of the cases were caused by structural abnormalities, while the other half by numerical. It is hypothesized that the difference in frequency may be due to epidemiological factors, hygiene conditions, chromosomal behaviors, etc.

All patients with classical and variants of TS, such as isochromosome X and mosaicism of Turner and isochromosome X, revealed that in females with PA, short stature appears to be the main clinical feature. The karyotype may play a significant role in the physical manifestations of TS patients, which is one explanation for this problem. The short stature homeobox (SHOX) gene in the pseudoautosomal region 1 (PAR1) is the major player in

this regard, and is thought to cause short stature in Turner syndrome. Haploinsufficiency of this gene results in growth failure (*Gürsoy et al., 2017*).

Insufficient estrogen production is another potential factor in Turner syndrome's short stature; many people also experience osteoporosis. This may result in a further reduction in height and a worsening of the spine's curvature (*Allahbadia & Allahbadia, 2016*). Isochromosome formation is a relatively frequent chromosomal aberration, mainly in X chromosomes. The cause of this phenomenon is the transverse division of the chromosomes, rather than along their length. The resulting isochromosomes have two short or two long arms, so they have an unbalanced chromosomal constitution, monosomy for the missing, and trisomy for the duplicated arms (*Gersen, 2013*).

Isochromosomes may occur during the anaphase of mitosis and meiosis through a mid-division of the centromere or U-type strand exchange. Turner patients with the Isochromosome X constitution appear only in short stature and primary amenorrhea, making it challenging diagnosis for physicians (*Michael, 2019*). Spontaneous menarche did not appear in the X long-arm isochromosome patients, even in mosaicism with other cell lines. The patient with isochromosome has high plasma gonadotropin (FSH, LH) levels with low estradiol and progesterone levels, much as in classical TS patients. The absence of a feedback mechanism might explain this hypertrophic hypogonadism phenomenon due to a lack of ovarian functions (*Soltani, Mirzaei & Ayatollahi, 2021*).

In this study, male karyotype [46, XY] was dementated in one case. This XY female had normal stature and structurally normal female external genitalia but an infantile uterus, while ovaries appeared tissue to scar. In this case, secondary sexual development did not occur at puberty, and gonadotropins ([FSH], [LH]) were elevated, and estrogen decreased.

The last case of abnormal karyotype revealed in this study was structural abnormalities in chromosomes 3 and 19 [46, XX,-3,+der3,-19, Del 19 p]. For the first defect, one copy of chromosome 3 was derivative shown abnormal in the q arm. The FOXL2 gene, located in the region (3q22.3), has a role in premature ovarian failure. FOXL2 gene is a forkhead transcription factor that is essential to proper female reproductive function and is central to ovarian development and maintenance (*Almukhtar et al., 2014*). Another defect, one copy of chromosome 19, was shown to be deleted in the p arm in all cells, which is a location for the AMH gene. This result implies an interpretation of the premature ovary failure, in which the AMH gene is responsible for producing a protein (Anti-Mullerian Hormone) involved in sex differentiation (*Almukhtar et al., 2014*).

Exon 10 encodes the C-terminal end of the extracellular domain, the entire transmembrane domain, and the intracellular domain of FSHR. Exon 10 is fundamental for signal transduction but is not necessary for ligand binding. The two most common SNPs in exon 10 are found at nucleotide positions 919 and 2039 (numbered according to the translation start codon with ATG as "1"), corresponding to amino acid positions 307 and 680, respectively, of the mature protein (*Wunsch, Sonntag & Simoni, 2007*) . Several research around the world have revealed on the rs6166 polymorphism and its relationship to a variety of disorders in the female reproductive system, including infertility. This polymorphism, which is located in the intracellular region of the receptor, has the ability to affect the glycosylation state of the intracellular portion and, as a result, the downstream
signaling response and receptor activity that occur when FSH binds to its receptor (*Nenonen et al., 2019*). The shift from A to G at position 2039 results in a change from Asparagine to Serine at position 680 in the receptor's intracellular domain, introducing a possible phosphorylation site. This modification, albeit containing only one base pair change, could impact the FSH receptor's downstream processes, affecting the menstrual cycle and resulting in the patient presenting with primary amenorrhea (*Casarini et al., 2014*). The single nucleotide polymorphism of the FSHR gene to amino acid positions 307 (rs6165) might consider as one of the susceptible factors that result in amenorrhea (*Nijeeb, Alkazaz & Yaseen, 2020*).

Many studies have investigated the principal cause of amenorrhea and the impact of mutation and polymorphisms on the disease. However, there is little knowledge about this in Iraq, so this study could help to seek the main genetic cause of amenorrhea and investigated the effect of FSHR gene polymorphism (Ala307Thr and Ser680Asn) with altered ovarian response in amenorrhoeic women.

The results of PCR-RFLP show the homozygous genotypes AA and GG of Ala307Thr (rs6165) in the FSHR gene to be the most common genotype in cases of PA; in SA, the homozygous genotype AA is more common. Furthermore, allele A is an etiological factor associated with the disease, while allele G is the protective agent in each case of PA and SA depending on the odds ratio.

The homozygous genotype GG and heterozygous genotype AG of Ser680Asn(rs6166) are more frequent in patients with PA and SA compared with healthy control women in the Iraqi population. Further, allele G revealed a highly significant positive association with the disease. In contrast, allele A was found to be negatively associated with the disease according to the odds ratio, and could thus be considered a protective allele.

Our study does not entirely agree with a previous study by *Achrekar et al. (2010)*, on Indian women; non-significant frequency distribution genotypes were revealed in Ala307Thr and Ser680Asn polymorphism in cases of PA and SA in comparison with the control group.

On the other hand, the study performed by *Nijeeb, Alkazaz & Yaseen (2020)*, on Ala307Thr polymorphism in Iraqi women with PA was in agreement with the significant frequency distribution of homozygous AA between PA and women with healthy menstruation genotype and allele A as a risk factor with amenorrhea (odds ratio greater than 1). Additionally, a study done by *Nijeeb, Shaker & Al-Ghanimi (2022)*, on Iraqi women, results of this study revealed that there is no association between FSHR gene polymorphisms with PA in comparison with healthy control women.

The present study emphasizes the need for cytogenetic analysis as an integral part of the diagnostic protocol which should be carried out to know the genetic basis of primary and secondary amenorrhea for precise identification of an etiology of chromosomal abnormalities and for appropriate management and counselling of these patients. The chromosomal abnormalities were most common in primary amenorrhea compared to secondary amenorrhea. Furthermore, numerical chromosomal abnormalities were the most common in this study, followed by structural abnormalities. Of these aberrations, the

Monosomy X chromosome/Turner Syndrome and its variants were the most chromosomal abnormality observed.

Variations in FSHR genes have an essential influence on ovarian function and can cause several defects of female fertility. The rs6165 and rs6166 gene polymorphism of FSHR are associated with female menstruation and can be used as a relevant molecular biomarker to identify the risk of amenorrhea in our population. After FSHR gene polymorphism, PCR-RFLP revealed that the homozygous genotypes AA and GG of Ala307Thr (rs6165) is the most common genotype in cases of PA. In SA, the homozygous genotype AA is more common. Furthermore, the allele A is appearing risky an etiological factor associated with the disease in term of odds ratio in both of PA and SA. The homozygous genotype GG and heterozygous genotype AG of Ser680Asn (rs6166) are more frequent in patients with PA and SA compared with healthy control women in the Iraqi population. Further, allele G revealed a highly significant positive association (risky factor) with the disease.

### Funding
The authors received no funding for this work.

### Competing Interests
The authors declare there are no competing interests.

### Author Contributions
- Bushra A. Kanaan conceived and designed the experiments, performed the experiments, analyzed the data, prepared figures and/or tables, and approved the final draft.
- Mushtak T.S. Al-Ouqaili conceived and designed the experiments, performed the experiments, analyzed the data, prepared figures and/or tables, authored or reviewed drafts of the article, and approved the final draft.
- Rafal M. Murshid analyzed the data, authored or reviewed drafts of the article, and approved the final draft.

### Human Ethics
The following information was supplied relating to ethical approvals (*i.e.*, approving body and any reference numbers):

According to the Helsinki Declaration, this study received approval from the University of Anbar's Medical Ethics Committee on February 13, 2022, in Ramadi, Iraq (approval number 8). All study participants provided their verbal or written informed consent

### Data Availability
The datasets generated and/or analyzed during the current study are available at NCBI: LC739718.1, LC739719.1, LC739720, LC739721, LC739722, LC739723, LC739724, LC739725.

## Supplemental Information

Supplemental information for this article can be found online at http://dx.doi.org/10.7717/peerj.15267#supplemental-information.

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
