# Peer review of "Cytogenetic screening of chromosomal abnormalities and genetic analysis of FSH receptor Ala307Thr and Ser680Asn genes in amenorrheic patients"

_PeerJ, doi:10.7717/peerj.15267_

## Round 0.1 · original submission · Minor Revisions

Dear Authors. Thank you very much for your submission. In reviewing the comments provided by the reviewers, I am requesting revisions before the manuscript can be considered further.

Reviewer 1 ·

Basic reporting

The article is well written which emphasizes on cytogenetic evaluation of patients with primary and secondary amenorrhea. Also the author has tried to look for polymorphic changes in FSHR gene in patients with normal karyotype and high FSH levels to find any association. The experimental design is well explained but the conclusion is not clear regarding the genotyping of FSHR gene polymorphisms.
What do you conclude after genotyping FSHR gene polymorphism?
Can rs6165 or rs6166 be considered as molecular marker for primary or secondary amenorrhea?
What is the impact of single nucleotide polymorphism (rs6166) of the Follicle Stimulating Hormone Receptor (FSHR) gene in females diagnosed with primary amenorrhea?

Experimental design

It is well designed.

Validity of the findings

Conclusion is vague regarding the genotyping of FSHR gene.
What is the impact of single nucleotide polymorphism (rs6166 and rs6165) of the Follicle Stimulating Hormone Receptor (FSHR) gene in females diagnosed with primary amenorrhea?
How does the knowledge of polymorphism affect the management of such patients?

·

Basic reporting

This manuscript by Kanaan and coworkers titled ‘Cytogenetic screening of chromosomal abnormalities and genetic analysis of FSH receptor Ala307Thr and Ser680Asn genes in amenorrheic patients’ describes karyotyping and FSHR mutation analysis in Iraqi women with primary and secondary amenorrhea. This is an excellent study and Sample is not high but it is neither small. The study design is appropriate for the research question. The introduction is written well, and the methods were clearly described. Results were clearly explained in the text and adequately supported by the figures and tables. The discussion is well written. However, the results and discussion sections require minor rearrangements.
I feel addressing the following issues would improve this manuscript.
1. Authors compared the results in this screen to the published literature. I appreciate the authors for doing this comparison. However, I ask the authors to shift this portion to the end of the results including the tables showing this comparison.
2. Please remove the figure and table headers from the results section. Please cite the tables and figures each and every time the test explains the results shown in the corresponding table/figure.
3. Gene name FSHR is missing in the introduction portion of the abstract.
4. Please write what frequencies of PA and SA are explained by the mutations and chromosomal abnormalities studied in this cohort. It would reveal what percentage of patients have none of these genetic aberrations. Using a flow chart to describe this would be more appealing.
5. Were chromosomal abnormalities and mutations screened in healthy subjects? Please describe in the results if they were screened. T would establish the baseline in the population.
6. I suggest authors to rewrite the sentences in lines 236-243. These sentences are complicated. Break them into multiple sentences to improve clarity.
7. Line 242: Do female study patients mean PA and SA together? Please be clear.

Experimental design

No comment

Validity of the findings

No comment

Reviewer 3 ·

Basic reporting

Readers of the PeerJ journal may find the article “Cytogenetic screening of chromosomal abnormalities and genetic analysis of FSH receptor Ala307Thr and Ser680Asn genes in amenorrheic patients” to be of interest. This study aims to establish the chromosomal and genetic changes in amenorrheic women.

Introduction :
1.The introduction is well-written and provides sufficient background on the study. In line 90, the authors stated that ‘”A mutant allele of these polymorphisms associated with the lower response of receptor and is significantly connected with the high level of basal FSH”. It would benefit the reader if more information on the implications of a high level of basal FSH is provided. May I know how a lower response of receptor is connected with high level of basal FSH ?
2. In line 95, the authors mentioned that the outcome of this study may detect the course of treatment or warning in patients for example in Turner syndrome. But it is well understood that turner syndrome is associated with early menopause and infertility. And usually, Turner syndrome is diagnosed before birth (prenatally), during infancy or in early childhood. Please clarify. The same goes for line 96, please clarify what do authors mean ‘ to help the clinician in announcing the recommendation to undertake hormone therapy?’

Experimental design

1. What is the basis for the number of patients recruited in this study? Is there any sample size calculation used?
2. The ethical statement is clearly stated in the manuscript and in the supplementary files.
3. Line 125. Hanging sentence. “After allowing it to clot in the water bath.”
4. Line 131, Blood sample for the chromosomal study. Is it for cytogenetic study/karyotyping as stated in line 137-138. If yes, I would like to suggest standardizing the use of these terms.
5. Line 147, please change to 100 µl. Similar corrections in line 152, line 155.Please standardize the use of units throughout the methodology section (example : minute or min)

Validity of the findings

Result:
1. Line 316 – Is there any description/explanation of the result in Figure 7 ?
2. Line 240 – 244, only mention the frequency and karyotype result. How about the rest of the results (hormonal status, ultrasonography findings, etc). Similar comment for table 4.

Discussion:
In my opinion, the discussion on the effect of FSHR gene polymorphism is insufficiently discussed.

Conclusion :
The author concluded that the study provides the need for cytogenetic analysis to understand the genetic basis of PA and SA (line 485-486). In my opinion, this conclusion does not reflect the results presented in the study and is very general.

Additional comments

No comment.

---

## Round 0.2 · accepted · Accept

Thank you for resubmitting the manuscript. The authors have sufficiently improved their manuscript in accordance with the reviewers' comments. In view of this, I recommend that the manuscript be accepted.

·

Basic reporting

Authors addressed my concerns. I have no further comments.

Experimental design

-

Validity of the findings

-

Reviewer 3 ·

Basic reporting

No comment

Experimental design

No comment

Validity of the findings

No comment

Additional comments

Line 433. remove , after Achrekar et al., (2010),
Line 450. Variations (capital V)